# Paleohydraulics and Complexity Theory: Perspectives on Self Organization of Ancient Societies

Charles R. Ortloff [1,2]

1    CFD Consultants International, Ltd., 18310 Southview Avenue, Los Gatos, CA 95033, USA; ortloff5@aol.com
2    Research Associate in Anthropology, University of Chicago, Chicago, IL 60637, USA

**Abstract:** Complexity theory provides a path toward understanding the development of ancient Andean societal progress from early settlements to later high population states. The use of modern hydraulic engineering methods to develop an understanding of the technical achievements of ancient societies (paleohydraulics), when combined with complexity theory, provides a path toward understanding the role of hydraulic engineering achievements to guide population increase and societal group cooperation on the path from early kin settlements to later statehood. An example case illustrating the paleohydraulics-complexity theory connection is presented for advancement of the pre-Columbian Bolivian Tiwanaku (600–1100 CE) society through their seasonal control of groundwater levels in urban city areas. This feature provided well water availability for city housing, public fountains, city hygienic and health benefits from the control of habitation dampness levels, water on a year-round basis for intra-city specialty crops, and the structural foundational stability of monumental religious structures. Commensurate with this application, Tiwanaku raised-field systems utilized groundwater control technologies to support multi-cropping agriculture to support growing population demands. Paleohydraulics theory together with complexity theory is applied to other major South American ancient societies (Caral, Tiwanaku, Chimú, Wari, Inka) to illustrate the influence of advanced hydraulic engineering technologies on advances from early origins to statehood.

**Keywords:** complexity theory; paleohydraulics; archaeology; societal development; water engineering; South American sites





## 1. Introduction

The 2023 Special Issue publication *CFD Investigation of the Water Supply and Distribution of Ancient Old and New World Archaeological Sites to Recover Ancient Water Engineering Discoveries* presented CFD results related to the hydraulic engineering discoveries of several ancient New and Old World societies; the present companion paper adds discussion of the relevance of several of these water engineering discoveries to the social evolution of ancient pre-Columbian societies of South America as derived from complexity theory perspectives.

The investigation of hydraulic engineering systems developed by ancient water engineers was examined with the use of CFD computer models and modern hydraulic engineering theory [1–8] to provide insight into the technologies used by their water engineers. In several of these references, modern water technologies used to examine UNESCO World Heritage sites in previous centuries brought forward new revelations with regard to the water engineering used by their hydraulic engineers for the first time. Remaining to be examined are the implications of the use of advanced water technologies on the socio-political, socio-economic, and socio-cultural aspects of ancient societies to develop higher levels of societal attainment. The present paper provides a path to address these issues using an application of complexity theory to investigate socio-political and socio-economic developments related to population growth and the societal advancement of pre-Columbian Andean societies. The logical structure of complexity theory's equation

systems provides the means to relate water engineering advancements to the social structure development of ancient societies. To illustrate complexity theory's use, the focus is on ancient societies of Pre-Columbian South America, as these societies' have a significant dependence on water engineering advances in order to survive the threatening ENSO El Niño and La Niña floods and droughts in coastal Peruvian ecologies, in addition to survival threats to societies in high altitude Andean mountainous environments. To supplement complexity theory's analysis predictions, the historical record of several Andean societies is detailed in order to examine the effects of climate change challenges.

Publications focused on this topic [1–16] include the societal structure and water supply and distribution systems of the Peruvian north coast Preceramic Caral (2500–1500 BCE) and the Late Intermediate Period Chimú society (800–1400 CE), the Middle Horizon Period sites of Andean Tiwanaku and Wari (600–1100 CE), and the Late Horizon Inka Andean sites (1400–1532 CE).

One source of ancient hydraulic engineering knowledge appears through lessons provided from nature observations translated into practical use for urban and agricultural water supply and distribution systems. One such lesson is that a stream originating from rainwater runoff over densely packed soil will carve a least-resistance cross-section channel shape to transport water at the highest flow rate possible for a given land slope. This observation provided an irrigation canal cross-section design used by ancient Chimú water engineers for several of their irrigation canals sourced from valley rivers such as those found in the field systems adjacent to the Chimú capital Chan Chan [2,5–7,17,18] on the north coast of Peru. Approximate half-hexagon cross-section, stone-lined canals are found in the Chimú Pampa Esperanza field systems north of the capital city of Chan Chan [3,5]; this canal design is the optimum straight-sided wall construction approximation to promote the low wall friction drag necessary to create high canal flow rates according to modern hydraulic engineering theory [19–21]. Further advanced Chimú hydraulic engineering is demonstrated from the ~900 CE Chicama-Moche Intervalley Canal construction, where the design water flow rate is made a near-critical maximum value by local canal cross-section shape changes as the lengthwise canal slope varies; this technology is close to modern hydraulic engineering practice [6,7,19–22]. The Moche Valley Chimú capital city of Chan Chan experienced extended drought in the ~1000 CE time period, as evidenced by Pampa Esperanza contracting canal cross section profiles [1] (pp. 25–26). This climate change effect effectively limited nearby canal supplied irrigation agriculture sourced by the Moche River and progressively lowered the groundwater level. To restore the water supply for urban and agricultural use, the 75 km long Chicama-Moche Intervalley Canal was constructed earlier to bring water from the high flow rate northern Chicama River to the adjacent southern Moche Valley agricultural field systems to increase agricultural production [1] (pp. 53–56) for Chan Chan's increasing population requirements. With field system water seepage into the aquifer from this supplemental water supply to augment the groundwater level, wells within several of the city's ten royal compounds had water availability resulting from the continued depth excavation to supply the royalty occupying the compounds. Supplemental to this drought restoration plan was the excavation of lengthy, wide, and deep subsurface troughs down to the water table located at high elevations between Chan Chan and the Pacific coastline to provide for additional crop growth from the moist trough bottom surface. As the ~1000 CE drought intensified over time with water table shrinkage to even lower depths, the original inland troughs were abandoned, and a new trough of similar proportions was constructed closer to the Pacific shoreline to intersect the declining water table to continue the agricultural production. These troughs were abandoned as the drought continued, with final troughs constructed even closer to the shoreline to intersect traces of the water table. This trough sequence is still observable from field observations after 1200 years of abandonment time. These comprehensive drought-defensive water system constructions indicate the Chan Chan's royal administration's knowledge of the canal supplied irrigation technology for surface agricultural field systems together with the water table hydraulic engineering applications necessary to sustain Chan Chan's population

through a long-term drought anomaly. The ancient water engineering involved is typical of a modern design approach and indicates a deep knowledge of the diverse levels of water technologies necessary for societal continuity during climate change challenges to survival. With the ENSO El Niño and La Niña water supply variations (mainly drought and flooding cycles), the Chimú coastal society had to develop advanced water supply technologies to sustain their population by the creation and development of advanced hydraulic engineering practices.

Further use of advanced hydraulic technology is exhibited at the Late Horizon Inka royal site of Tipón located close to the Inka capital of Cuzco, where a water supply open channel undergoes a sudden major width reduction to create a critical flow in the narrower downstream channel supplying water to a display waterfall. This design feature creates critical flow [19,21] in the contracted downstream channel section that limits disturbance effects (channel bends, width changes and vortex flows in downstream basins) from propagating upstream to destroying flow symmetry in four waterfall spillage channels [2] (pp. 69–95), and preserves flow aesthetics, as would be expected for a compound occupied by Inka royalty. As many Inka urban, royal and administrative palace sites have elaborate aqueducts, reservoirs and channel water distribution systems [21–25], it is expected that future excavation activity will reveal further details of the engineering knowledge base equivalent to that shown at the Tipón site.

A further demonstration of advanced Andean hydraulic engineering knowledge relates to the raised-field agricultural system and urban water supply system of the Bolivian (600–1100 CE) Tiwanaku [1–3]. Here, the raised fields consist of vast areas of raised agricultural berms separated by swales excavated down to the groundwater level. The water level in the swales was maintained at a constant level in the dry season by a system of canals from a nearby river to supply water to berm plant root systems supporting agriculture. In the rainy season, drainage canals returned the excess water back into the Tiwanaku River at a downstream location to maintain swale water height. The net result of this water management was double-cropping through seasonal changes that was necessary to support Tiwanaku's 20,000–30,000 urban population. At the Tiwanaku city center, the water supply from the mountain-based springs was guided by channels into a deep 'moat' encircling the elite residence city area with Akapana, Putini, and Kalkasasaya Temple religious and administrative housing monuments [23]. In the rainy season, runoff water channeled into the moat infiltrated the groundwater aquifer to maintain the water table height constant by drainage channels carrying excessive moat runoff and seepage water to the nearby Tiwanaku River. In the dry season, perpetual springs distant from the urban center supplied channeled water into the moat to keep the water table depth constant [2] (pp. 1–30). With the water table maintained close to the urban ground surface on a year-round basis, specialty llama and alpaca pasturage areas within the city were maintained together with the control of dampness levels within city housing structures for hygienic and living quarter mold prevention health purposes. A further advantage of the maintained constant depth water table was the structural stability of the massive Akapana Pyramid and Kalasasaya structures within the moat water that were maintained at a constant level through seasonal changes in rainfall availability. Here, the water saturation of the monument's compacted aquifer underlying these structures prevented [2] structural deformation. Groundwater height control provided the continuous tappable water source for wells observed from excavations in urban housing structures [26] (p. 165). Together, the water control systems used for urban and agricultural sustenance provided high living standards for the 20,000–30,000 city population during seasonal changes in rainfall [2,8,9,14,26,27]. Combined with pastoral alpaca and llama protein food sources together with Lake Titicaca marine food sources and food imports from satellite towns [28], Tiwanaku city residents had a secure food supply base that was largely immune to transient climate change effects. A further indication of the water technology is evident in agricultural field systems south of urban Tiwanaku. Two adjacent canals exist a few meters apart from each other: one negative low slope canal takes water from the Tiwanaku

River to supply raised-field systems during the dry season, while the other canal with an exceptionally low positive slope extracts excess water from the raised-field system during the rainy season to maintain swale water height. To construct side-by-side adjacent canals with opposing slopes (one positive; the other negative) indicates that Tiwanaku water engineers were capable of slope measurements on the order of ~$10^{-4}$ degrees, which is comparable to Roman accuracies [2] (p. 89). Of further note, raised-field agriculture was sustained through freezing altiplano (~3500 m altitude) temperatures by the capture of radiative solar heating of the raised-field swale water by several degrees centigrade over the sub-zero freezing temperature. This water temperature increase was aided by darkened swale decomposed bottom plant material enhancing the capture of heat within swale water channels between planting berms [8]. The IR radiative water heating of swale water prevented ice formation, as the latent heat withdrawal necessary for freezing was not available, resulting in the survivability of hardy crops in the altiplano winter season. These and other hydraulic engineering discoveries are a sampling of the water control technologies underlying the agricultural and urban water supply base of major Andean cities that played a key role in their societal development. Historically, innovative water supply and distribution technologies emerge as solutions to existing ENSO climate change challenges and characterize innovative societies that reimagine modified present water technologies to make them better for future generations.

Technical hydraulic engineering accomplishments discovered through use of CFD and modern hydraulic analysis at many other Old and New World sites are presently known [1–4]. It is clear that hydraulic engineering advances supporting population growth by sustainable agriculture influence the advancement of living standards of many different societies and provide the means of advancement from rural townships to cities, where different creative elements of society in close communication can mutually provide the benefits of a high standard of communal living.

The evolution of a society based upon technical achievements in water management requires an administrative structure to manage and oversee the use of water technologies that are vital for societal continuity and growth. The degree of societal evolution and its accompanying administrative structure involves the management of land, water, labor and technology resources specific to different Andean ecological environments where different societal systems and water availability and supply systems exist. For several of the early Andean Aymara kingdoms (such as the Pukara, Chiripa, Taraco, and Alto Ramirez) from the ancestral . . . to the Tiwanaku, moiety administrative systems composed of two complementary units with differences in ritual, political and economic status existed to influence the later Tiwanaku societal structure composed of a major capital city and surrounding settlements of Lukurmata, Pachiri and Khonko Wankané [10,26–28]. For the Inka governance of its provinces from Ecuador to mid-Chile, ~40,000 multi-level administrators responsible for the control of diverse province populations from Ecuador to mid-Chile conformed to central authority dictates originating in Cuzco [24], the highland capital of the Inka empire. For the north coastal Peruvian Chimú, nine separate royal adobe wall compounds related to each successive ruler's governance period existed, with ruler successions originating from dual corporate organization [15]. Such elaborate administrative architectural structures are features of the top-down resource control necessary for societal advancement. The discussion of the importance of technical advances overseen by elite administrative water management as related to social structure development are summarized by many authors in the literature of ancient South American societies [1–3,10] (pp. 364–366); [23] (pp. 23–25); [29–37], but questions remain as to the specifics of how ancient Andean societies put into practice their water technological knowledge to influence their cultural patterns, their socio-economic and socio-political security base, and their food supply base, and how water technology advancements influenced their societal development from small communities to city status. For the Peruvian north coast Chimú state where complex hydraulic technology was demonstrated in river-supplied, intra-and intervalley irrigation canal designs, the earlier lower population Moche (300–800 CE) society occupying similar

coastal areas exhibited lesser levels of hydraulic knowledge for limited agricultural areas used by their lower population level. Clearly, the question of population size and the food supply base to support it, as made possible by the available water engineering technology, was a vital consideration in societal advancement. The Late Intermediate Period Chimú developed certainty about food resource production for their large population that required a higher level of technical inventiveness to provide irrigation water over vastly increased agricultural areas. For Andean highland societies, Lane demonstrated the use of dams to trap and conserve rainfall water supplies during seasonal changes in water availability, with cooperation between local communities in charge of the construction and maintenance of local water resources. Diverse types of water engineering structures and management systems thus appear in different Andean societies due to differences in local ecologies, differences in socio-political, socio-economic and socio-cultural societal structures, as well as from different responses to ENSO climate variations that affected societal continuity. Given the unpredictability of nature's climate and weather variations, and their effect on agriculture and societal sustainability, nature as a benevolent guide may seem doubtful as a reliable path that yields benefits to human society given periods in Andean history characterized by extensive droughts and catastrophic El Niño flood events. What is common to many advanced Andean societies, however, is the will of individuals to create a management system to govern the use of labor for cooperative rather than individual group land management of agricultural systems. This step requires the creation of a leadership class to guide the collective action of the population toward societal change with economic benefits for all. This class existence is demonstrated by major administrative structures found in most Andean societies that had achieved statehood status. An obstacle to societal cooperation vital to coordinating activities toward a positive, universally accepted goal originates from challenges to universal acceptance by segments of the populace to leadership dictates. This path holds for societies in their formation stages, as well as for societies already established by a family succession of rulers. For both cases, acceptance by the populace is key to their success and continuance, as many water control projects require the enlistment of vast labor resources together with the logistics to manage support facilities: all requiring the cooperation of the masses for a successful outcome. Given that different Andean societies evolved different management and administrative structures, the path toward cooperation for major infrastructure projects likely involved agreements between groups with different interests, religious and social structures, and different historical backgrounds and viewpoints as to how infrastructure improvement projects should be carried out; how cooperation toward the creation of a society sharing common goals is achieved is a subject within the realm of complexity theory analysis to investigate. To examine the creation of a managerial class that governs its populace in a cooperative manner given diverse cultural pattern differences in different Andean stratified societies, complexity theory provides a methodology to examine the interaction factors behind societal structural changes and development.

## 2. Societal Development According to Modern Complexity Theory

To understand the mechanism behind societal cooperative efforts, complexity theory [36] provides insight into how technical advances can be the catalytic impetus behind strategic decisions that arise from alternative strategies that lead to societal advancement. Provided that the attractiveness of a particular strategic decision advanced by elite management or individual groups can be translated into economic and labor saving benefits for individuals, cooperation within the populace to willingly perform tasks involving cooperative labor investment from all society members is assured. This process involves discussions between individuals and groups where advocates of different points of view, not necessarily those of elite management, vie for acceptance for their point of view. When the majority of a population accepts a proposed concept that coincides with official policy and customs (or vice versa), then the means for a universal cooperative effort from all society members is assured. The steps towards this end involve dictates that are accepted by

the populace (or accepted by passive non-response) and are not necessarily ordered by elite decree but accepted by most of the population given the economic assurance and lifestyle benefits to individual society members. While many discussions on societal evolution give catalytic socio-political, socio-economic, and socio-cultural events as drivers to change [37], complexity theory examines leadership factions within societal groups espousing different solutions to problems that promote beneficial change. This is done in present-day democratic societies through elections, with prominent leaders espousing different political philosophies, but in many ancient societies, some form of collective agreement promoted by influential leaders influenced societal change based upon the promotion of new visions of achievable economic progress that are acceptable to all.

In historical reality, nonaccepting $\beta$ groups opposing a leadership $\alpha$ strategy group continue to revise their strategy to overtake $\alpha$ leadership by either peaceful or confrontational means. Proposed solutions have time limits, as new $\alpha$ and $\beta$ proposals are rapidly advanced in response in an effort to garner acceptance from the general populace. For this result to happen, acceptance by the populace must incorporate a benefit change $A_\alpha(t)$ and a strategy change $\alpha(t)$ evident to the general populace promoting group acceptance change. Complexity theory formulation provides a governing equation, Equation (1), that, upon solution, provides the time evolution path toward understanding how gradual changes occur in a society commensurate with technical advancements that have labor saving and economic benefits shared by all rather than just by an elite group. Solutions to Equation (1) can represent paths for time-dependent societal change or solutions for evolved changes from one strategy to another strategy. From a more comprehensive viewpoint, the decision processes involved in the creation of a cooperative union rely on economic advantages apparent to all; this is stated by Durant [38] (pp. 51–57) as " . . . there is a contest among individuals, groups, classes and states for food, materials and economic power that are rooted in economic realities- in such cases, motives of leaders may be economic, but the result is largely determined by the passions of the masses." From this statement, there is a political reality familiar to those who study past and present history describable by complexity theory that investigates the management path toward creating decisions accepted by the masses that is key to societal change (for better or for worse).

For water engineering technologies of pre-Columbian societies applied to agriculture and urban use, several of which are described in the Introduction, different experiments and insights put into practice with the evaluation of results over time provide an effective means to their incorporation to update earlier systems. The role of a leadership class to guide or enable decisions [39,40] is noted as a principal factor in the advance of societal development. Given incomplete and preliminary details about the structure of Andean societies' management and decision-making capability that resides within elite (or non-elite) groups to guide technology-based decisions, the origins of societal progress from this source is subjective, given that different societies' hydraulic science knowledge depth can only be assayed from the analysis of several existing archaeological remains of their water systems. However, some degree of self-organization leading to progress, whether originating from a top-down elite management group or from bottom-up localized groups, must prevail in order to develop the water control systems for urban and agricultural use, as noted from the archaeological record. The choices of different adopted agricultural strategies are based upon the benefits evident to all participants in terms of decreased labor leading to increased agricultural productivity. As an agricultural surplus is essential to the growth and continuity of a society, particularly for formative Andean societies emerging from allyu and multi-generation community roots [12], it is of interest to examine societal complexity development from a complexity theory perspective based on the selection of an adopted agricultural strategy among various choices put forward by different elements of society. The selected strategy forms the bonding structure of shared interests to move forward on rules and practices accepted by all agreeing society members. This goal, once achieved, overcomes contrary population elements conditioned by class inequality, notions of superiority among different kin groups, the newness of the urbanism experience, the

uncertainty of economic realities, male/female visions of what is important, differences in religious and behavioral standards, previous sociopolitical trajectories, and the usual vagaries of human nature that at times go against practical realities. In many Andean historical cases, a non-egalitarian society managing agricultural resources was the preferred outcome, and projects were conceived and executed by a common consensus without elite class management. For the present discussion, a consensus path toward the incorporation of water engineering advances that benefit all society members that promotes the development of an elite management structure is examined by complexity theory. This proceeds through individuals leading societal subgroups endeavoring to convince members of competing groups to join them to form a collective that shares similar goals. For established hierarchies, continuance in power depends upon their ability to retain favor with their population by succeeding in overcoming interior challenges from both interior and exterior sources as well as those from nature in the form of major climate change challenges.

To explore self-organizing societies and those already established by royal decree using complexity theory, for an agricultural strategy $\alpha$, then $A_\alpha$ represents the benefits of choice, and $\alpha$ and $X_\alpha$ represents the number of people having adopted strategy $\alpha$ at a given time. The $\alpha$ strategy contains two factors: $\alpha_1$, the convincing power of $\alpha$ group leaders based upon the promotion of economic advantage to all society members, and $\alpha_2$, the technology advance in water engineering underlying the conversion of groups to $\alpha$ perspectives based upon labor-saving technical, health and economic benefits to dissenting $X_\beta$ group participants. Sample $\alpha_2$ technologies related to hydraulic engineering applied to advances in urban and agricultural systems are given in the Introduction for ancient New World societies, for example. The fractional number of individuals $X_\beta$ considering an alternate strategy $\beta$ with benefits $A_\beta$ out of the total population (N) is then $X_\beta/(X_\alpha + X_\beta)$. In practical terms, $X_\alpha$ may represent an elite class population imposing its will on non-conforming $X_\beta$ subgroups. Routledge [41] (p. 27) expresses this division as "how sovereign power is carried out against all odds and against the material interests of at least some of those involved", a route by which $X_\alpha$ can dominate $X_\beta$ opposition. While this situation characterizes many historical examples [42], a trend toward ultimate acceptance of an $A_\alpha$ strategy based upon demonstrated successful economic and societal stability advantages for all promoted by an $X_\alpha$ elite minority can convince the $X_\beta$ minority to accept elite management control. In terms of complexity theory, those individuals wishing to leave strategy $\beta$ in favor of strategy $\alpha$ will be proportional to the number of $X_\beta$ people multiplied by the relative attractiveness of strategy $\alpha$, which is $A_\alpha/(A_\alpha + A_\beta)$. As $X_\alpha + X_\beta = N$ (the total population), then for the recruitment rate (a) per individual solicitor and t = time, the differential equation governing the rate of change of the $X_\alpha$ population favoring $\alpha$ strategy follows that of Nicolis and Prigogine [36] (p. 239),

$$dX_\alpha/dt = a\, X_\alpha\, \{[A_\alpha\, X_\beta/(A_\alpha + A_\beta)] - [A_\beta\, X_\alpha/(A_\alpha + A_\beta)]\} \tag{1}$$

or, after $X_\alpha + X_\beta = N$ substitution,

$$dX_\alpha/dt = a\, X_\alpha\, [N\, A_\alpha/(A_\alpha + A_\beta) - X_\alpha] \tag{2}$$

This equation gives the rate of change of the number of people selecting option $\alpha$. Note that the economic benefits derived from $A_\alpha$ water technology improvements that increase agricultural production and minimize labor input increase the supportive $X_\alpha$ population, as well as help convert the other $X_\beta$ working classes not directly involved in agricultural production. For cases where the attractiveness of a particular strategy is evident over time due to positive economic results enjoyed by all classes of society, i.e., when $A_\alpha$ is large compared to $A_\beta$, then a greater number of the population participates in its use, as is evident by the larger values of $dX_\alpha/dt$. Here, $0 < A_\alpha < 1$, $0 < A_\beta < 1$.

Nondimensionalizing Equation (3) with $X_\alpha/N = X_\alpha'$ and $X_\beta/N = X_\beta'$, then with $t/t_0 = t'$, then solving Equation (2) after dropping the ' notation,

$$X_\alpha(t) \sim e^{\Omega a\,t}/(e^{\Omega a\,t} - 1) \tag{3}$$

where $\Omega = [1 - (A_\beta/A_\alpha + A_\beta)]$. This long-term $t \gg 0$ solution gives the effect of a favorable agricultural and/or water engineering advance $A_\alpha$ strategy to increase the population $X_\alpha$ that endorses the $\alpha$ strategy over a long period of time. From Equation (3), $t \gg 0$, $X_\alpha(t) \to 1$ (by L'Hospital's Theorem), indicating total dominance of strategy $\alpha$ by the majority $X_\alpha$ population. Here, $X_\alpha = 1$ denotes the total acceptance of strategy $\alpha$ by the population.

For an example case from the Tiwanaku archaeological record, the population increase supporting the $A_\alpha$ strategy was directed by the elite management as benefitting the urban population over increasing time results, as it produces economic benefits for all. The food supply security at Tiwanaku proper would be instrumental to induce other population centers in the Moquegua Valley, coastal Chile Azapa, and southeastern Cochabamba, as well as sites flanking the western edge of Lake Titicaca [10,26,43], to adopt the benefits of Tiwanaku's Bolivian capital's $\alpha$ strategy, which would induce trade and cooperation between the Tiwanaku capital and the satellite towns [32]. This inducement to share the beneficial agricultural and cultural benefits of association with Tiwanaku proper would then expand the influence of all things Tiwanaku and lay the foundation for the creation of the Tiwanaku empire (as noted from the archaeological record). From Equation (3), the population segment $X_\alpha$ endorsing $A_\alpha$ benefits dominates any minority $X_\beta$ opposition, as cooperation (compared to coercion) takes time to accomplish, given the success of the recruitment rate (a) from the increasing $X_\alpha$ population segment to overcome the opposition of $X_\beta$ individuals. Satellite towns and villages share the leadership and benefits of the elite Bolivian central Tiwanaku center to form an expansive state noted as $\Omega < 1$ societies. The recruitment rate (a) from the central authority that peacefully projects the benefits of incorporation into the empire implies the existence of a Tiwanaku leadership that convincingly influences large segments of their city and satellite populations; this is opposed to command conformance by edict, such as would be expected from a dominant hierarchical management structure that is not responsive to opposition to their rule.

If a decision is made by an autocratic minority to enforce strategy $\alpha$ with benefits $A_\alpha$, then $\Omega \to 1$ as $A_\beta \to 0$ and the conversion to a total $X_\alpha$ society is complete, as Equation (3) predicts. This then represents a step forward from rural village life to a coordinated population of different mutually supportive work classes coming together in a city structure, all supporting a common vision of economic benefits derived from common interests. An interpretation of Equation (3) is that when a successful, agricultural system exists together with a progressive technical advance $\alpha_2$ and common interest $\alpha_1$ cultural strategies, then the top-down acceptance ($X_\alpha \to 1$, $X_\beta \to 0$) of a mutually beneficial situation for all participating labor groups occurs and a managerial group acceptable to all society members is put in charge of implementing the successful $\alpha$ strategy with the population growth assured by adequate food supplies. Individual allyu kin groups possessing the ability to generate a successful agricultural strategy $A_\alpha$ using local control only has also been demonstrated for Tiwanaku in its early formative stage [44]. This was followed by elite management governance in later stages [26], where communal labor was evidenced in major construction projects, communal festive bonding feasts, advanced agricultural practice, and military expansion policies—all in a city environment. City formation is then a natural occurrence, as different agricultural workers and labor specialty groups working in cooperative union are most efficiently productive when in a close contact city communication environment. While elements of society may not initially conform to changes from established norms that are promoted by the leadership, the provided economic benefits of a more secure existence prevails, and then community consensus ultimately prevails when society members agree that they benefit from new policies [33] and close cooperation. Further research [32,40,44,45] emphasized collective ritual behavior in state societies as a common bonding $A_\alpha$ strategy element of self-organization. In this

respect, paccha and huaca rituals and festive ceremonies [2] (pp. 6–12), [10,23,46] are additional key bonding elements observed in Andean societies that are best served in a city environment. Provided that leadership guiding the acceptance of an $A_\alpha$ strategy with mutual economic and social benefits is apparent to a majority body of the population, then, as Equation (3) indicates, elements of a collective, cooperative society appear in city form as a preliminary step toward statehood and empire when neighboring satellite societies adopt similar rules of existence to those of the main governing center city. Note that while knowledgeable agricultural management leads to sustainable population increases, geometric population increases with linear food supply over time leads to Malthusian extinction; thus continuous agricultural surplus and the water engineering technology to support it is necessary to sustain population growth. Thus emphasis on hydraulic technology as it is related to agricultural production is a critical factor in Andean societal development. A further option arises in times of crisis, where communities yield to absolutist forms of leadership authority [18,40,41] (p. 17). In cases of extended drought or flooding emergencies inducing the collapse of agricultural systems, populations cede total control to authority figures that provide the basis for collective action to organize labor to begin the restoration of damaged water supply and distribution systems. In such cases, a new $\beta$ strategy with an $A_\beta$ solution is the only option with an $X_\beta$ group under immediate control that may displace a former governance $X_\alpha$ officialdom. From a solution of Equation (2), $X_\alpha \sim e^{-a\,A_\beta\,t}$, indicating the decline of the former $X_\alpha$ leadership class and its outdated $\alpha$ strategy under an emergency condition affecting agricultural production as $A_\beta$ declines. In certain cases, the existing $X_\alpha$ leadership may utilize features of earlier agricultural increase projects to maintain their leadership role in drought crisis times to restore a semblance of agricultural production and restoration to continue their leadership role. The Chicama-Moche Intervalley Canal originated by the Chimú leadership is a prime example [6,7] of this leadership strategy, as severe drought affecting the agricultural fields of the capital city of Chan Chan mandates mandatory enlistment of the population to serve as laborers to construct this new water supply system to reinvigorate the capital's agricultural base in record time to eliminate the population collapse in the capital city.

Outside of hierarchical governance societies lie societies that resist the establishment of hierarchical leadership and proceed with organized labor that resist the emergence of local leaders [47]. For this case, there is only one $A_\alpha$ strategy, and $X_\alpha$ remains fixed and not subject to change in time. From Equation (3), for a society that defers to egalitarian rule and prefers community consensus government, for t >> 0, only $X_\beta$ exists, confirming the coherent society rule by consensus rather than by an $X\alpha$ elite class.

Equation (1) is also applicable to pre-state societies characterized by a clan hierarchy without hereditary rank where community, rather than individuals or elite groups, provide focus for community rituals and the initial materialization of a leadership class. This social structure is found in Late Preceramic societies of the Peruvian north coast and central highlands, where complex architecture and agricultural systems begin to appear that are derived from community cooperation [48] (pp. 45–47), [49]. In Late Preceramic and Formative sites, (3000–1500 BCE), the site of Caral in particular, the trade of marine resources available from coastal fishing communities for agricultural products available from inland sites becomes the start of a more complex societal system relying upon irrigation technologies and enlarged agricultural field systems under a central management, as trade comes with traders versed in commercial dealings [8,16,50]. From the central administrative center at Caral, $X_\alpha$ leadership controlled eighteen other Supe Valley sites sharing the same $\alpha$ and $A_\alpha$ values by trading resources specific to each site to compose a version of a cooperative pre-state society. With the evolution of early societies, complex architectural structures originated, indicating the creation of religious and administrative centers as formal steps necessary for later statehood. The initial reliance by individual Tiwanaku allyu clan groups exploiting lakeside surface field plots later translates into the cooperative labor reliance of many groups that effectively manage raised-field agriculture over a vast scale (~19,000 hectares). This efficiency improvement is based upon $\alpha_2$ groundwater control

refinements related to spring-sourced canals and drainage channels to control seasonal raised-field swale height in order to preserve agricultural output as well as changes in berm width patterns to extract more agricultural product per unit raised-field land area [3]. The implementation and installation of technical improvements relies on the recruitment of large labor resources guided by a centralized, top-down control that additionally managed planting and harvesting cycles [10,26]. Thus, a single agricultural $\alpha$ strategy guided by a central authority guided Tiwanaku's large city population effectively as an outgrowth of reliance on competing individual kin groups, with individual strategies only accountable to their individual welfare. For the Inka, established elite management $X_\alpha$ controlled all aspects of land resettlement, labor assignments, and land allocation for populations under their control [22,23]. Population conformance to Inka hierarchical control was achieved through proficiency in the technical aspects of different ecology agricultural systems together with multi-level personnel control enforced by state controlled kuracas governing the agricultural resources of the expansive Inka empire [23].

As the archaeological record shows for later stages of major Andean societies, vast arrays of ceremonial structures and corporate administrative compounds exist within large urban compounds, signaling the arrival of top-down decision making authority consistent with the $\Omega \rightarrow 1$ strategy. Archaeological theory, as applied to early ancient Peruvian and Bolivian societies, proves difficult with regard to the extraction of populace attitude details of societal change over time, but alternatively relies on the building of large ceremonial and administrative complexes requiring cooperative group infrastructure to complete, as the proof of societal advances through emergence of an elite rulership class takes place. Was there an initial form of authoritative coercion involved in the decision processes ($\Omega < 1$) involving the accession by societal leaders to organize *corveé* labor resources according to Equation (2), or did a $\Omega \rightarrow 1$ governing hierarchical structure originate from democratic consensus, or from the assumption of power by a hierarchical elite to rapidly stimulate the efficient progress towards a more efficient use of land, water, labor, and technology resources? As administrative, religious, ceremonial, and elite class compounds separate from secular classes remain in the archaeological record as a source of interpretation of societal structure that created them, many questions remain unanswered with regard to the transition process of early Andean societal structures. The results of complexity theory indicate at least one path toward consensus and cooperation through the recognition and acceptance of economic benefits that can be experienced by all society members, a logic which in any society has the strength of consensus building.

The examination of Inka society (1400–1532 CE), for which many accounts are available from Spanish chroniclers related to their social, economic, and political organization, indicate a well-ordered and well-organized society with many rules, rites, and rituals accepted as orthodoxy by a population controlled by a top-down Inka hierarchy. Other earlier Andean societies, such as the Chimú, Wari and Tiwanaku, followed this path with ultimate acceptance of the stability, prosperity and predictability that top-down management provided. As populations increased in these societies over time, the archaeological record shows that top-down management was vital for organizing labor intensive projects in the form of water resource management for agriculture and urban use, as well as religion-based monumental structures. While societal structure variations exist in the path toward empire, many of the major Andean societies that are dependent upon agricultural success appear to follow a variation of Wittfogel's vision of a hydraulic society [51–53].

## 3. From Theory to Reality

In discussions to follow that probe the source and progression of the Andean path to statehood and empire, the case is made for the conversion from bottom-up to top-down management strategies based upon economic and labor-saving improvements of a population that are mutually shared between different labor groups in a society. While only two strategies ($\alpha_1$ and $\alpha_2$) are considered for the present discussion, the further extension to more competing strategies and strategy variants involving different supporting segments

of the population are summarized in Nicolls and Prigogine [36], (p. 240, Equation 6.15) by the extension of Equation (1) to multiple clan and kin groups $X_i$ supporting different $A_i$ strategies with different success levels of recruitment strategies. Although the idea of systematic planning with anticipated results leads to the development of planners within a top-down societal management structure, a further progress path derives from a trial agricultural system that produces marginal benefits over time and is then replaced from lessons learned by a new system whose $A_i$ benefits are only apparent over an extensive trial run period under stable climate conditions. This progress path requires a stable environment over time to demonstrate benefits over a previously used agricultural system. Given nature's climate and weather vagaries, this learning path is dependent upon lessons from prior experimental trials which may not have time to demonstrate a replacement agricultural system. Thus, emergency trial innovative solutions may proceed from new climatical environmental and ecological conditions such as long term drought and periodic El Niño flooding affecting the agricultural and social structure environment. The creation of the Chicama-Moche Intervalley Canal that was designed to bring water from the Chicama River to the later post-drought period reinvigoration of the desiccated Moche Valley Chimú agricultural lands is a prime example of an emergency solution requiring a massive labor input controlled by the top-down Chan Chan royal administration to preserve the seat of government power and the surrounding agricultural resources supporting the capital city. Under stable climate conditions, there is time to derive agricultural surplus progress from sequential agricultural experiments to induce top-down societal management structure's planning and execution of a reliable system that is based upon the knowledge gained from past experiments. Under drastic climate change, this state is replaced by mandatory survival projects, lest civilization collapse. The Andean archaeological record is replete with examples of societies that manage through technological innovation to survive climate change events; the record is also replete with less innovative civilizations that were overwhelmed by the magnitude of a climate change event that disappear from the archaeological record. The hydraulic technology available to the Tiwanaku, the Chimú and other major Andean societies served as the basic element to progress to a more complex social structure capable of more complex projects serving their population with improved economic and security benefits.

　　　Excavations of Tiwanaku raised-field agricultural areas west of Lake Titicaca at two meter depth revealed traces of an earlier raised-field design that had shorter water channel lengths between narrower planting berms than later, more productive raised-field designs. This $\alpha_2$ advantageous berm wavelength and swale width change made over centuries derived from trial raised-field configuration differences over time that improved agricultural productivity; this an example of trial-and-error advances designed to gain knowledge in a stable climate period. The emergence of an organized agricultural system over long time periods generates a population increase (as Equation (3) predicts), with a top-down management center emphasizing religious practices designed to influence the deities' oversight of the progress of a society. Significant technical advances in hydraulic and hydrological sciences are observed in Tiwanaku, Wari, Chimú and Inka archaeological history that are indicative of an active branch of government focused on higher levels of technical improvement. As Andean history is characterized by several infrequent, but long-lasting, unstable climate periods [2,9,14,49,54–56], many defensive strategies evolved to protect agricultural continuity; these strategies resulted from the memory of past destructive climate events and the innovative defensive agricultural design solutions that evolved and were incorporated into later agricultural systems of major Andean societies.

## 4. Tiwanaku Societal Collapse: ENSO Drought According to Complexity Theory Models

　　　From Tiwanaku social organization [8,10,12,14,26], an initial allyu kin societal structure prevailed by using early versions of raised-field agriculture. The consolidation of allyu groups led to the reorganization of raised-field systems on a larger, more integrated

scale that promoted increased agricultural productivity with lower labor input. This suggests that a cooperative agricultural strategy advantage $A_\alpha$ predominated over previous strategies so that the $X_\alpha$ majority conversion prevailed. As demands of a more secure food supply and provision for increasing population became evident ($\Omega < 1$), a higher level of management expertise prevailed ($\Omega \to 1$) to expand raised-field productivity. As Tiwanaku hierarchical rule managed more complex water projects involving urban ground-water control [4] (pp. 97–103) and used labor organizations to carry forward complex water control projects, consensus from allyu kin groups was achieved, resulting in a higher level of economic security for Tiwanaku society. From the 10th to post 11th CE centuries, Tiwanaku city abandonment gradually occurred as the population decreased due to the long-term drought-induced contraction of agricultural capacity [9,51–56]. The decrease in population size is represented by $N \sim K \exp\{-kt^2\}$, where K is the initial population size, k is the rate of progression of the drought, and (a) is the society participation level. For $\Lambda = a\,K\,A_\alpha/(A_\alpha + A_\beta)$, Equation (2) becomes:

$$dX_\alpha/dt = \Lambda\,X_\alpha\,\exp\{-kt^2\} - a\,X_\alpha{}^2 \qquad (4)$$

where $\Lambda = a\,K\,A_\alpha/(A_\alpha + A_\beta)$. The solution to Equation (4) for the population decline of the ruling elite $X_\alpha(t)$ is:

$$X_\alpha(t) = \exp\{\Omega\,\mathrm{erf}(t)\}/\{(a/k^{1/2})\int_0^t \exp\{\Omega\,\mathrm{erf}\,(t)\}\,dt + [X_\alpha(0)]^{-1}\} \qquad (5)$$

where $\Omega = (\Lambda/2)\,(\pi/k)^{1/2}$ and where $X_\alpha(0) > 0$ and $X_\alpha(0)$ is the population at the start of the precipitous drought at time $t = 0$. The erf(t) term is the Error Function given by:

$$\mathrm{erf}(t) = (4/\pi)^{1/2}\int_0^t \exp\left(-s^2\right)\,ds \qquad (6)$$

At $t = 0$, the initial condition $X_\alpha(0)$ is recovered on both sides of Equation (5). The $X_\alpha(t)$ population consists of the ruling class of society; the $X_\beta$ population is the nonsecular working trade class segment, with both classes sharing a similar fate, as climate conditions affecting agricultural production led to population decline. The solution shown in Figure 1 from Equation (5) indicates the decline of the $X_\alpha(t)$ ruling over time for k and K values representing 10th–11th CE century extended drought conditions. The governing leadership class $X_\alpha(t)$ segment diminishes in a damaging way as the disappearance of leadership elites disassembles guidance figures directing the fate of the population. Rituals and celebratory ceremonies were no longer performed, and trust in leadership's authority was compromised.

Under these conditions, social fragmentation and political unrest were followed by the abandonment of trust in deities and their connection to the elite class [57]. From on-site observation, an excavated deity statuary was defaced and a deity representation carved into a stone block was found as part of a building wall with the deity face reversed into the interior part of the wall during the extended drought period. The occurrence of human sacrifices in the main areas of Tiwanaku in this period was noted [10,26] as offerings to deities designed to restore stability. Figure 1 provides the numerical evaluation of Equation (5) for K = 1, k = 1, a = 1 and $\Omega = \pi^{1/2}/2$. Drought timing and intensity is regulated by the increased values of K and k; the a = 1 notation denotes the full involvement of all members of the population that were subject to drought conditions.

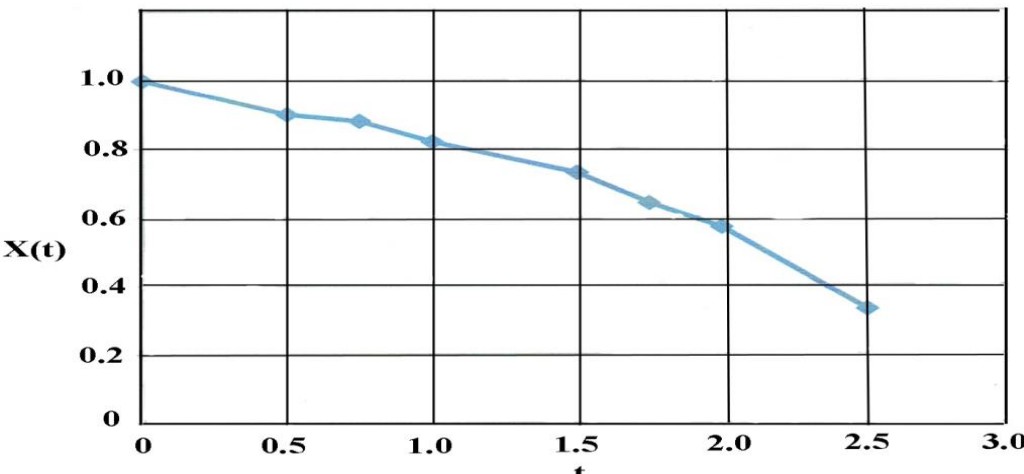

**Figure 1.** Typical decline in ruling class $X_\alpha(t)$ with time t (in years) from Equation (5) using typical K and k values that prescribe the drought intensity and duration.

According to the archaeological record of the Tiwanaku society, the population decline was accompanied by the dispersal of small kin groups abandoning the city to areas with localized water resources to permit limited forms of cocha agriculture. The lure of city life with the advantage of unity and prosperity derived from sharing common goals together with managed land areas that guaranteed adequate food supply supplemented by trade from satellite societies occupying different ecological realms provided the initial draw for rural populations to initially band together to form a city. As the later period drought intensified, the advantages of city life receded as the stable food supply diminished and key management personnel were no longer available to make leadership decisions to manage labor and resource allocation effectively. Social fragmentation and political discontent followed with the abandonment of formerly beneficial deities and rituals that were no longer effective in maintaining societal harmony [57]. The abandonment of near-lake-edge raised-fields and the transfer of farming to more distant areas from the Titicaca Lake edge proceeded as the distant water table remained sufficiently high to support the vestiges of raised-field agriculture and to limit open-field farming [11]. The decrease in Lake Titicaca's height from extended drought [13] affected near-lake raised-fields directly, as the groundwater profile declined with the lake surface shrinkage and decline. Rainfall infiltration over previous centuries into plains near mountainous areas distant from Lake Titicaca transferred groundwater vital for cocha farming to portions of the urban area. The presence of cocha farming pits excavated down to the declined water table supporting lower population groups was noted, as the drought intensified consistent with the dispersal of the urban population into smaller groups [10] (p. 266). In the final stages of drought, the Tiwanaku urban center became abandoned by ~1100 CE, and the later return of rainfall norms formed the basis of the next phase of population development, which was characterized by multiple, dispersed defensive sites that guarded precious water resources.

## 5. Visions of the Last Days of the City of Tiwanaku

The final stages of urban Tiwanaku during the extended drought are described [58,59], and further details of the social structure disassembly indicated from complexity theory usage derived from the solution of Equation (5) is shown in Figure 1. The establishment of extended drought conditions that led to the gradual demise of urban Tiwanaku and its raised-field agricultural systems [3,9,13,14] is substantiated by geophysical data originating from Thompson's ice core data [54–56].

Figure 2 summarizes the drought initiation and duration by integration of the nine-year moving averages of the Cordilla Negra Quelccaya mountain ice cap thickness data [56], where the sequential deposited ice layer thinness from drilling core data denotes rainfall

decline. The extensive drought from ~1000 to 1300 CE slowly lowered water table levels in raised-field swales, promoting the loss of crops to freezing events together with water unavailability to plant root systems [8,14]. This extended drought likewise affected the demise of the contemporary highland Peruvian Wari society, as well as influencing the contraction of canal irrigation systems of the north coastal Peru Chimú society [3] (p. 253) indicating its influence over large parts of both Peru and Bolivia over several centuries. Given the different geophysical ecologies and the different levels of susceptibility to drought of major Andean societies in the 1000–1300 CE time period, and the different defensive responses to preserve their agricultural base, some societies, including the Tiwanaku and Wari, vanished from the archaeological record, while others, particularly the Chimú, managed to sustain their existence by the resourceful use of alternate water sources provided by the use of the Chicama-Moche Intervalley Canal [6,7].

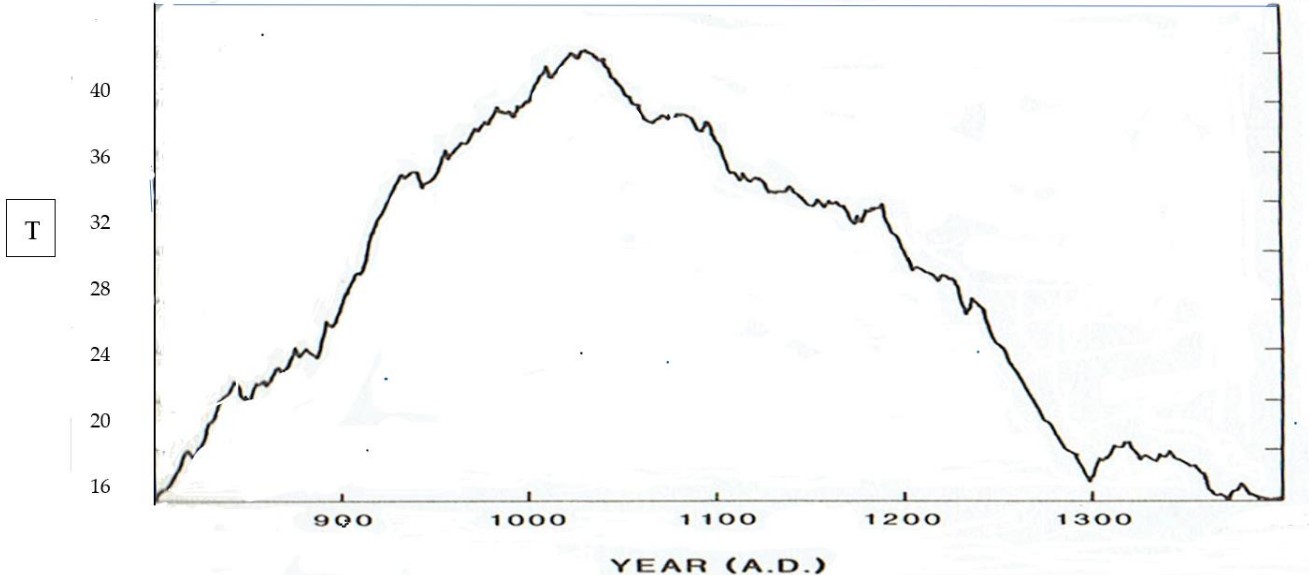

**Figure 2.** The moving average of Huascaran Mountain's ice core yearly deposit layer thickness begins a decline at ~1000 CE, indicating the initiation of severe drought conditions. The ordinate scale T is in centimeters.

Information related to the last days of Tiwanaku city life is available from multiple stable isotope methods involving the analysis of skeletal remains dating from the ~1100 AD time-period, which corresponds to the city abandonment dates [58,59]. The dietary change from previous norms was experienced by the city population as the drought intensified; these changes included the absence of fish from the diet, and there are no reported instances of child remains that contained fish or marine food sources which might be indicative of a change in the Lake Titicaca marine resource base to supply a protein source to the city population.

As Tiwanaku's working class population decreased with the loss of skilled fishermen as well as the $X_\alpha$ management direction (Figure 1), the population decline and outward migration from urban Tiwanaku continued with the loss of specialized industries' key members, together with population numbers sufficient to continue with the sowing and harvesting of food resources.

The decline in fish and marine resources available to citizens may represent the allocation of high nutrient marine food types to the most productive members of society that remained, with the hope that their skills may be available to restore elements of the previous way of post-drought life; alternatively, fish species resource depletion may result from higher salinity, as the Lake Titicaca lake level declines, limiting fish catch amounts as well as nutrient values. The results from [60,61] indicate the presence of maize as a food source in the ~1000 CE time period, indicating the importation from different Tiwanaku

satellite areas where maize could be raised and transported in dried kernel form. Recent studies [58–61] detailed the societal collapse of contemporary Middle Horizon Wari sites due to food shortages in the same time period leading to the collapse of that society, the capital of which was centered in the Ayacucho region of Peru. Figure 3 indicates that severe drought conditions present in Peruvian and Bolivian highland societies as well as in Peruvian north coastal areas [3] (pp. 24–26) at ~1000 CE, together with an earlier ~600–700 CE ~30-year drought period on the north coast of Peru, altered the survival fate of several Andean societies, confirming the extent and duration of the drought on societal survival and extinction [62]. In some cases, societies survived by altering their farming methodologies (or by the conquest of other societies with significant land and water resources), while other societies disappeared from the archaeological record [2,3,59,60].

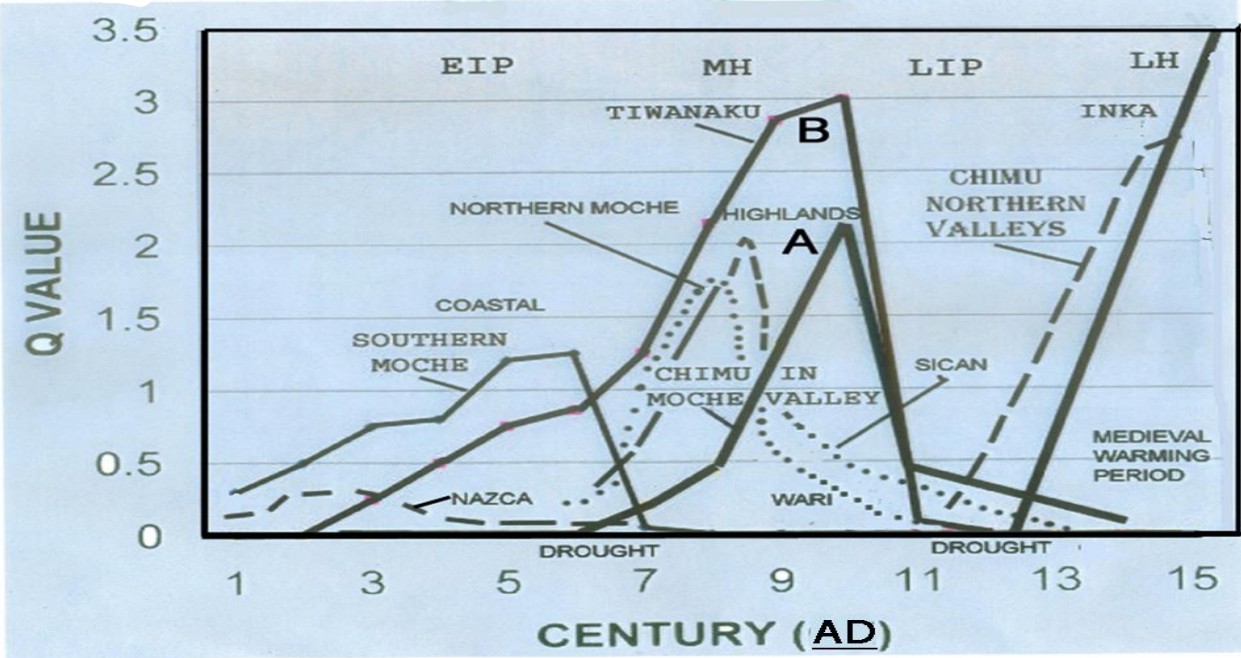

**Figure 3.** Drought led to societal decline for both the Tiwanaku and Wari polities as well as influencing the Peruvian north coast Moche, Sican and Chimú Societies (Ortloff and Moseley 2009). The Q Value ordinate denotes the survivability value of a civilization as defined in [18].

Of interest in Figure 3 is the post drought recovery in the 13th–14th century CE that led to Inka expansion and control over vast areas of coastal and highland western South America, with no state level polities from other societies left to contest their dominance. For the Late Horizon

For the Inka, the conquest of Peru by Pizarro's conquistadors in 1532 basically eliminated the earlier Inka ~100 year dominance of Pacific coast and inland sites of South America from (present-day) mid-Columbia to mid-Chile regions, and was subsequently replaced by Spanish administrative control of these areas. Based upon the research of the effects of the on the demise (or continuation) of many Peruvian and Bolivian empires [18] from Early Horizon to late Horizon times, several late Intermediate Period droughts lasting several hundred years were characterized by continuous small percentage declines in rainfall levels that had different effects on different sites' survival depending upon the ecology of their agricultural system types. For the Bolivian altiplano highland site of Tiwanaku, drought manifested itself as a slow decline in the ground water level, affecting raised field dependent agriculture [3,9,14]; for Peruvian Chimú coastal sites dependent upon rainfall runoff from the Cordillera Blanca mountain areas to rivers supplying canal based irrigation systems, drought had a more severe effect over a shorter period of time [1,7] and limited agricultural production, as Figure 3 indicates.

## 6. Complexity Theory and Societal Evolution

Returning to the complexity theory analysis of societal change, Equation (1) is based on only two competing groups: $X_\alpha$ and $X_\beta$. Given human nature, new challenges and sociopolitical changes in a society alter previous norms and produce many additional competing group factions: $X_\alpha, X_\beta, X_\gamma, X_\delta, \ldots$ that vie for dominance by a single group. Equation (1) can be modified to accommodate additional $X_i$ factions, promoting different strategies whose success in winning converts relies on the persuasive powers of group leaders to achieve converts from other groups. Given human nature, the proposed collective group economic benefits are sometimes not perceived by some as individual member benefits, which leads to subgroups choosing not to retain former group identities and practices. With respect to Tiwanaku society, Goldstein's interpretation [28] of the evolved societal structure as determined from the formative processes that led to state development is given as " . . . the dynamic interplay among myriad counterposed factions in both state and periphery representing different factions interacting to produce behavioral procedures that define their society". This statement is consistent with complexity theory models that allow for group competition through the specification of their $X_i$, $A_i$ and $\alpha_i$ terms that may challenge the ruling and other societal classes that resist challenges to their continuance. Andean history demonstrates cases where climate change affecting agriculture was viewed as a deity's rejection of the leadership, with subsequent modifications of the authority of the leadership class [9,57].

## 7. Early Versions on the Optimum Structure of a Society: Perspectives from Complexity Theory

Perspectives on societal structure development have a history that was understood by scholars from previous centuries. The discussions of the optimum structure of a society from philosophers of the Greek golden age, particularly discussions between Plato and Aristotle [63] (pp. 331–332), relate the views of land-apportioned individuals enlisted to, and committed to, voluntary communal labor to achieve community goals; this was expressed as "friend's property is common property" according to Plato's *Ideal State* from his *Laws*. Aristotle's modification (from his *Politics*) of Plato's propositions involves private ownership of property with the proviso that "friend's goods are common goods" means that the results of individual labor (primarily for agricultural products) are available to one's neighbors upon request. Imbedded in Aristotle's view is the consideration that all workers will not (or cannot) contribute equal amounts of voluntary labor but will share an equal amount of the agricultural production. Another view from Epicurus' *Rules for Living* would have a "state run by wise men" as the ideal state. Further research [63,64] traced the conditions of state success and the accompanying organization of power necessary for this to be achieved in Greek city states: their research on this topic proceeds from Greece's Golden Age under Pericles, where the involvement and approval of the citizenry was vital for government practices. For Roman and Hellenistic societies, the importance of water for urban and agricultural use is key to societal advancement [65–72]. A later view from the turmoil of the 17th century [65,66] saw the state and its rule of law as the antidote to "war of all against all", and the "fear of violent death at the hands of another" as the way to peaceful coexistence between enemies. The crucial role the state plays in the domestication of humanity is foremost in Hobbes' *Leviathan* or *The Matter, Form and Power of a Common-Wealth Ecclesiastical and Civil Society*. Further thoughts on the role of the city as a civilizing entity was expressed in later centuries, referring to the Roman presence in the British Isles [67] as " . . . to live well instead of merely living, was the membership in an actual physical city". From thoughts and writings about how societies could be ideally composed and the leadership necessary to achieve this end, history has demonstrated much experimentation along these lines, with various degrees of success for equality and individual freedoms being integral to state formation. Scholars over the centuries have developed many theoretical paths to explain societal development. These include the theory of evolution (society proceeds in a definite direction to a higher levels); unilinear

evolution theories (all societies pass through similar successive stages to the same end); and multilinear evolution (changes occur in several ways but do not progress in the same direction). Later scholars, such as Childe [68], laid down ten traits (the presence of cities, labor specialization, agricultural surplus, monumental architecture, a ruling elite, science, writing, standardized artwork, long distance trade, and solidarity based on residence rather than kinship) that define state formation, although all of these elements need not be present for the transition from farmer collectives to chiefdom-level societies. Childe's emphasis on a processional viewpoint was that archaeology is a branch of anthropology and that the conditions of societal development result from the creation of material items that lead to surpluses from agricultural production, tradable export products, complex irrigation networks, and implements the supporting production that was once concentrated in leadership elites, and that this formed the basis for societal advances. Childe's views accepted that archaeologists delivered 'cultures' that were based upon a material culture were influenced by Marxist views that material things were more important than ideas for societal advancement, and that social advancements derive from material conditions with modes of production that produced equal economic benefits for all. Childe's view was that Marxist diffusionism functioned in a society where humans were not inventive or inclined to cultural evolution, and thus were susceptible to cultural change through economic benefits equally available to all society members. The Marxist perspective involves class struggle to achieve its goals; this contrasts with Childe's views of a material culture evolving naturally without class struggle. With respect to complexity theory trends as described for pre-Columbian South American societies, Childe's Marxist materialist-influenced perspectives appear relevant, as much in the way of the advancement of their societies depended upon invention and the use of advanced irrigation technologies ($a_2$) to deliver the agricultural surplus necessary for population growth, together with binding religious and social rituals ($a_1$).

Liverani [69] (p. 6) concluded that state formation was not inevitable from early kinship groups, but progress relied on an agricultural surplus that was "not for consumption within the family but for the construction of infrastructures and for the support of specialists and administrators, the very authors of the social revolution itself". Progress supporting this transition is based on the communal use of agricultural surplus and the creation of an administrative class supporting and dedicated to the creation of a state's infrastructure; this view parallels complexity theory's equation systems predictions that rely on the creation of a vibrant agricultural base and leadership formation for Andean societies that is likely applicable to other early world societies. Progress toward statehood from complexity theory perspectives involves strategy cooperation on all matters between diverse groups to achieve social harmony and progress. Complexity theory thus provides the path whereby advances in technology and social cooperation rituals lead to the creation of a governing leadership elite class that concentrates the derived surplus from trade and agricultural resources to direct the advance from early diverse kin group settlements to city status. This path formally proceeds from complexity theory, requiring the cooperation of society members to be governed by commonly accepted rules. From Equation (3), $X_\alpha (t) \sim e^{\Omega a\, t}/(e^{\Omega a\, t} - 1)$, society evolves when different kin groups coalesce into united cooperative $X_\alpha$ members whose growth in time ($X_\alpha \to 1$) over increasing time t, Equation (3)) depends upon $a = a_1 + a_2$, where $a_1$ is composed of shared social rituals and strategies (religious practices, social binding feasts, expansionist policies, shared behavioral beliefs, agreed upon leadership rulers) and $a_2$ is composed of hydraulic engineering advances that provide food supply security and surpluses for societal security, as well as for advanced urban living standards.

From Greek and Roman authors and for pre-Columbian societies, societal advances emphasize both $a_1$ and particularly $a_2$ considerations given the importance of water for urban and agricultural use. From Wittfogel's viewpoint [51–53], hydraulic engineering advances in dynastic Chinese eras provided the basis for societal stability, as they provided defenses against destructive Spring season flood events [2] (pp. 340–349) as well

as providing innovative irrigation networks to support large populations. Wittfogel's emphasis on water technologies created by dynastic Chinese emperors to forge their empires derived from their absolute rule that controlled vast labor resources from the obedient working classes This is but one means to characterize progress toward statehood and empire creation in Chinese dynastic times, but is not applicable to other societies that have different paths to statehood.

The examination of the archaeological record of the societal structure of the Bolivian Tiwanaku society from its elementary kin group structure to the later state level appears to follow elements of Childe's and Liverani's precepts. This follows from the agricultural surplus provided by hydrological science advances using raised-field agriculture that led to the creation of an elite governing class, as predicted by complexity theory. Other authors [30,31,70,71] emphasize scientific and engineering material observations and advances at world city sites which were previously kept as seemingly unrelated and independent bits of information, only to be later shown to relate to each other through an encompassing theory that tied together the previously scattered information bits into "a close and effective relationship." This path of key technical and engineering developments is vital to the evolution of modern western science [30] (pp. 224–254), given many examples from 18th century Enlightenment Period origins [71], and is a key provision described by complexity theory's Equations (1) and (5) that incorporate $\alpha_2$ considerations. In a similar manner, the sophisticated hydraulic technologies exhibited by Andean societies [1–5] followed a similar evolutionary path as hydraulic engineering technology proceeded through cumulative incremental verification steps before their use to advance stable societal and agricultural progress. This path is consistent with complexity theory's 'A and $\alpha$' conditions that are specified in Equation (1). From complexity theory, the process of social formation that permits different views to compete for dominance ($A_\alpha$, $A_\beta$, ... ) by different competing elements of the population ($X_\alpha$, $X_\beta$, ... ) defines the culture change that emerges. This model characterized early Tiwanaku allyu kin groups that were deciding on more efficient ways to maximize agricultural production by collective labor enlistment that raised the economic status of all participating members of different kin groups. This procedure produced natural leaders that formalized their evolution to elite status, and whose importance increased as continued success resulted from the economics of the labor projects that they conceived and managed. In summary, complexity theory provides a logical explanation path, through its governing equations and solutions, to explain and understand the formation over time of advanced world societies from elementary levels characterized by the evolution of a leadership class utilizing a wide spectrum of engineering advances to promote high living standard levels to their populations.

## 8. Conclusions

The use of complexity theory provides a formal means to examine the basis of societal progress in Andean (and other) societies, and is new to archaeological and anthropological analysis. Equation (1) relates the expansion of major ancient Andean societies to hydraulic and hydrological science that maximized agricultural production and improved urban living standards; this path is but one among many that contribute to societal development. With the benefits of the increase in agricultural productivity, societal sustainability and security led to the development of a managerial class designed to oversee and develop society. From different elements of society with different capabilities, ambitions and viewpoints, effective elite management could enlist labor from all societal classes to cooperate in industrial scale agricultural projects that provided economic benefits for all, as predicted by complexity theory formalism.

Over ~4000 years, from the early Preceramic to Late Horizon times, Andean societies' use of advanced water supplies and distribution technologies provided one foundational element for the self-organization into hierarchical management polities, as noted from the archaeological record. Complexity theory thus provides a formal path to describe how different elements of societies working together in cooperative union can produce a

path to higher levels of societal development, in addition to the creation of an overseeing managerial class to direct progress toward this end. Complexity theory thus has a universal application with regard to understanding the progress and evolution of societies given the different challenges and constraints that come with historical evolution.

**Funding:** This research received no external funding.

**Data Availability Statement:** All data contained in this manuscript is available and can be shared with the public.

**Conflicts of Interest:** The author declares no conflict of interest.

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
