# Peer review of "Paleohydraulics and Complexity Theory: Perspectives on Self Organization of Ancient Societies"

_water, doi:10.3390/w15112071_

Round 1

Reviewer 1 Report

The paper references both Wittfogel and Childe, two researchers with shared intellectual roots in Marxism who developed in very different directions. One or two paragraphs might be added summarizing the main points of the Wittfogel thesis (with which not all 21st century readers will be familiar) and highlight how the Complexity Theory approach diverges from Wittfogel's technological determinism and Childe's stages of societal evolution.

Author Response

In response to your review, additions have been made to expand Chapter 7 by adding, as you requested, discussion on Childe-Marx influence, Childes' steps toward social evolution, discussion of Wittfogel's theories, added discussions of Childe's works, and new comments on the relation of past author's CFD publications to Complexity Theory results. This added discussion ties together Complexity Theory results that are relevant to the works of Chapter 7 authors.  An additional reference [72] has been added to add to the discussions of Hellenistic authors as well as some further comments related to Figure 3.Thank you for your review as changes made add further detail tio the manuscript!

Reviewer 2 Report

The article addresses the issue of complexity theory at a crossroads with the study of palaeohydrology and social organisation.

The article is quite innovative. We consider the literature review to be quite complete. The bibliographical references are adequate and up to date.

The paper presents a good accomplished internal structure. The methodology applied is coherent and effective.

The conclusions are consistent with the evidence and arguments presented.

We leave just one note that we think could help make the article more effective for the reader. The article gained a lot from the introduction of maps showing the location of the geographical areas referred to in the text. Also, some images that illustrate the archaeological and heritage remains of the hydraulic structures and/or water engineering technologies would be welcome. 

Author Response

Thank you for your review. The manuscript has some additional comments that add more to Chapter 7 and Figure 3 results. A referral to the earlier paper on CFD results has been added to the start of the Introduction as requested. This to relate the prior CFD studies paper to the effect on societal evolution as given in the present paper using Complexity Theory.

Reviewer 3 Report

Dividing the (now long) paragraphs into shorter entities would help

the fluidity of the narrative for the reader.

Many of the descriptions of the water-distribution systems would

greatly benefit from accompanying illustrations.

Equation (1) is presented rather far after it is first discussed.

There are a lot of (very minor) details with punctuation, spacing, and

mathematical notation that need cleaning up pre-publication.

I strongly advise against using the capital and lower-case versions of

the same letter (in this case K) to denote separate concepts, as this

is prone to create confusion when people discuss the work. Maybe one

could be a greek letter like kappa?

In general, the work is well written and I found it delightful to read.

Figure 2 lacks an (explicit) axis label for the vertical axis.

For some reason, the font changes near the end of page 14 and then

changes back in the second paragraph of page 15.

Figure 3 would be better without a background color. 

Author Response

The Figure 3 axis label has been added. The font change effect you mentioned has been corrected. Some addition text has been added in Chapter 7 to expand the relation of Complexity Theory to the work of earlier authors. Addition discussion has been provided for Figure 3. Thanks for your review!
